# Realistic Full-Body Motion Generation from Sparse Tracking with State Space Model

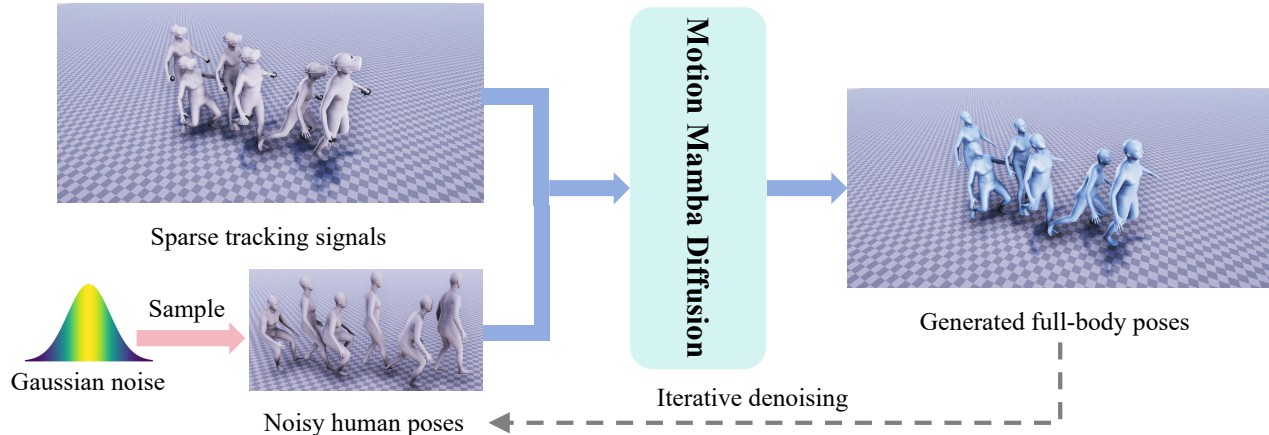

**Figure 1: Overview of the denoising process. Taking sparse tracking signals and noisy human poses as inputs, our method generates clean full-body poses through iterative denoising.**

## ABSTRACT

In the domain of generative multimedia and interactive experiences, generating realistic and accurate full-body poses from sparse tracking is crucial for many real-world applications, while achieving sequence modeling and efficient motion generation remains challenging. Recently, state space models (SSMs) with efficient hardware-aware designs (i.e., Mamba) have shown great potential for sequence modeling, particularly in temporal contexts. However, processing motion data is still challenging for SSMs. Specifically, the sparsity of input conditions makes motion generation an ill-posed problem. Moreover, the complex structure of the human body further complicates this task. To address these issues, we present Motion Mamba Diffusion (MMD), a novel conditional diffusion model, which effectively utilizes the sequence modeling capability of SSMs and the robust generation ability of diffusion models to track full-body poses accurately. In particular, we design a bidirectional Temporal Mamba Module (TMM) to model motion sequence. Additionally, a Spatial Mamba Module (SMM) is further proposed for feature enhancement within a single frame. Extensive experiments on the large motion capture dataset (AMASS) demonstrate that our proposed approach outperforms the latest methods in terms of accuracy and smoothness and achieves new state-of-the-art performance. Moreover, our approach runs in real-time, making it ideal for employment in practical applications. The source code will be made public upon acceptance of this paper.

## CCS CONCEPTS

• **Computing methodologies** → **Motion processing**; **Neural networks**; *Tracking*.

## KEYWORDS

Human Motion Generation, Sparse Tracking, Mamba, Diffusion

## 1 INTRODUCTION

With the recent popularity of generative multimedia and interactive experiences, tracking full-body human movement has become a highly demanded feature. However, there are no strong tracking signals for the entire human body in the typical AR/VR scenarios. Common VR systems are usually composed by Head Mounted Displays (HMD) and handheld controllers, in which Inertial Measurement Unit (IMU) sensors are utilized to track head and hands. While these devices can provide resourceful upper body motion information, the lower body is not tracked and must be estimated from the limited information provided by the upper body joints. The ill-posed intrinsic nature makes generating realistic full-body motion particularly challenging.

Various methods have been proposed for this task, many of which are based on the Transformer architecture [46]. While Transformer models [6, 11, 21, 28, 38, 48, 57, 59] are renowned for their ability to capture long-range dependencies within a sequence, there remains a concern regarding their accurate understanding of temporal motion relationships among frames, primarily due to their

*ACM MM, 2024, Melbourne, Australia*

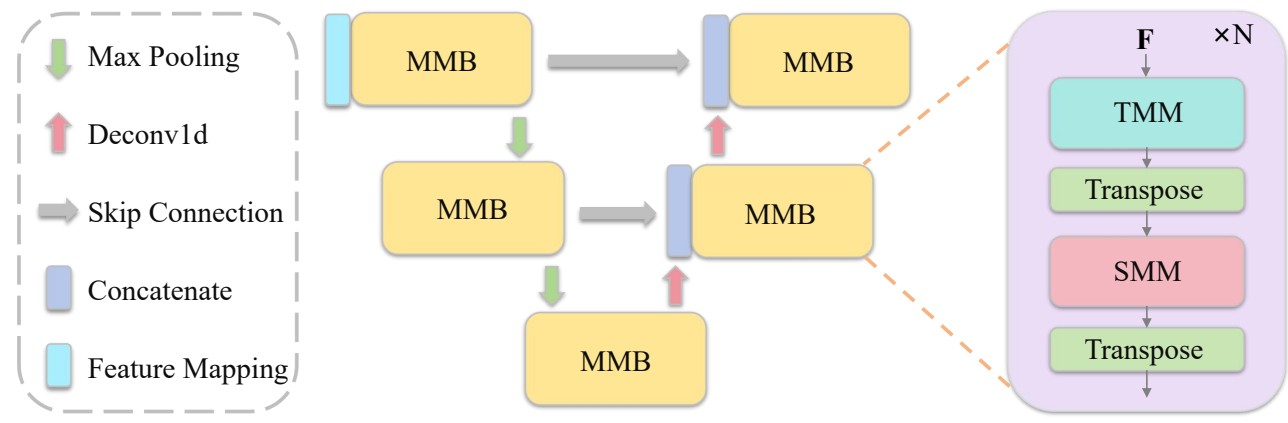

**Figure 2: The framework of the proposed Motion Mamba Diffusion (MMD). It starts with Feature Mapping to map the noisy human poses and sparse tracking signals to the unified latent space for subsequent process. Then a U-shaped architecture is applied to process motion information across different timescales. In particular, we utilize the Motion Mamba Block (MMB) to construct it, which consists of two key components tailored for motion generation, i.e. Temporal Mamba Module (TMM) and Spatial Mamba Module (SMM). Specifically, TMM focuses on capturing the temporal dependencies inherent in motion sequences, while SMM is employed for feature enhancement within each frame.**

dependency on positional encoding. A recent work, AGRoL[12], employs a MLP-based network for this task. Nevertheless, its reliance on 1D convolution to capture temporal relationships within motion sequences may limit its modeling capacity, thereby leading to insufficient performance. Recent advancements in State Space Models (SSMs) [17, 18, 32, 40, 47], notably Mamba [14], have demonstrated significant promise in sequence modeling with selective scanning mechanism, indicating a promising direction for building motion generation models based on this approach. However, it is not trivial to employ SSMs in this task. Generating human movements with high accuracy is of vital importance, yet the sparse nature of input conditions means this task is actually an ill-posed problem. Furthermore, the intricate structure of the human body also makes it very difficult to generate smooth and realistic movements.

In light of these issues, we propose the Motion Mamba Diffusion (MMD), a novel conditional diffusion model, which generates human poses through iterative denoising, as illustrated in Figure 1. It effectively leverages the robust generation capabilities of diffusion models in conjunction with the sequence modeling proficiency of SSMs. As shown in Figure 2, the proposed MMD adopts a U-shaped architecture to process motion information across different timescales, and the Motion Mamba Block (MMB) is designed to construct it, comprising two key components tailored for motion generation. First, a bidirectional Temporal Mamba Module (TMM) is employed for motion sequence modeling of sparse inputs. This module excels at identifying temporal dependencies among frames, facilitating a thorough comprehension of the dynamics inherited in motion sequences. Second, we apply a Spatial Mamba Module (SMM) to enhance the motion features within a single frame. This module is dedicated to improving the understanding of the human body structure within each frame.

The proposed MMD remarkably advances the performance of motion generation, notably improving the key metrics like MPJRE, MPJPE, and MPJVE by 13.53%, 14.55%, and 6.83%, respectively,

compared to the previous state-of-the-art method AGRoL [12] on AMASS [31]. Overall, the contributions of this paper are summarized as follows:

- A SSM-based framework, termed Motion Mamba Diffusion (MMD), is proposed for motion generation. It is a pioneering method for full-body pose estimation from sparse tracking signals with the incorporation of the selective scanning mechanism and denoising diffusion model.
- Two innovative components tailored for motion generation, Temporal Mamba Module (TMM) and Spatial Mamba Module (SMM), are introduced to construct the proposed MMD. Specifically, TMM is employed for motion sequence modeling of sparse inputs, while SMM is utilized to enhance the motion features within a single frame.
- Extensive experiments have been conducted to evaluate the proposed method, and the experimental results on the widely-used AMASS benchmark illustrate that our approach outperforms the state-of-the-art in terms of accuracy and smoothness. Further analysis reveals the contribution of each component to the performance improvement. Additionally, MMD runs in real-time, providing a crucial advancement for creating realistic virtual avatars in real-world applications.

## 2 RELATED WORK

### 2.1 Motion Generation from Sparse Tracking

Full-body motion generation from sparse tracking signals of body joints has attracted significant interest in the research community. Many prior works, such as [23, 52], have utilized up to 6 body-worn inertial sensors, typically distributed over the head, arms, pelvis, and legs. However, in many practical Head-Mounted Display (HMD) settings, only 3 tracking signals are available: the head and 2 wrists. CoolMoves [2] was the first to estimate full-body motion using only 3 tracking signals from headphones and handheld controllers.

Nevertheless, the adopted KNN-based method interpolates poses from a smaller dataset with specific motion, raising concerns about its scalability to larger datasets with diverse subjects and activities. Recently, various learning-based methods are proposed for this task. For instance, AvatarPoser [24] employed a Transformer encoder to extract deep features from the input signals and decouples global motion to guide pose estimation. AGRoL [12] introduced an MLP-based architecture for motion reconstruction. Other methods approached sparse input body tracking as a synthesis problem. To that extent, VAE-HMD [10] proposed a Variational AutoEncoder (VAE) architecture, while FLAG [3] opted for a flow-based method.

## 2.2 Denosing Diffusion Models

Emerging from the realm of image generation, denoising diffusion models [7, 13, 22, 27, 33, 43, 56] have garnered considerable attention due to their capability to yield high-quality results and adeptness at handling large datasets. These models acquire a probabilistic understanding over a denoising process applied to inputs, gradually refining Gaussian noise to the desired output. Furthermore, diffusion models excel in supporting conditional generation. For instance, ILVR [5] guided the generative process to generate superior-quality images based on a provided reference image. In a similar vein, GLIDE [35] explored diffusion models for text-conditional image synthesis. Furthermore, UniDiffuser [4] empowers a single model to accomplish multiple generation tasks with high quality. Recent advancements have extended diffusion models to motion synthesis. However, these endeavors have primarily focused on the text-to-motion [30, 39, 53] or audio-to-motion [8, 25, 51]tasks, with scant attention given to tracking the full body from sparse inputs. AGRoL [12] introduced the first diffusion model solely for motion reconstruction from sparse inputs. Nonetheless, there remains a concern regarding its precise comprehension of temporal sequence modeling, primarily stemming from the adopted MLP-based architecture. Our approach, Motion Mamba Diffusion, innovatively addresses this limitation, thereby leading to more accurate and smoother motion generation.

## 2.3 State Space Models

Recently, State Space Models (SSMs) are introduced into deep learning [44] as state space transforming. Inspired by continuous state space models in control systems and benefiting from HiPPO initialization [15], LSSL [18] demonstrates the potential to tackle long-range dependency problems. However, due to the prohibitive computation and memory demands, LSSL is practically unfeasible. To resolve this issue, S4 [17] proposes parameter normalization into a diagonal structure. Subsequently, various variants have emerged, encompassing complex-diagonal structures [16, 19], diagonal decomposition coupled with low-rank operations [20], and selection mechanisms [14]. These models are further integrated into larger representation models. Amid these advancements, Mamba [14] introduces the Selective Scan Space State Sequential Model (S6) Block, representing a promising innovation for addressing long sequence modeling across diverse tasks. Recent work has introduced Mamba into the field of computer vision [37, 42, 50, 54, 58], while [55] presents a text-to-motion framework.

## 3 PRELIMINARIES

State Space Models (SSMs), especially with the advancements made by structured state space sequence models (S4) [17] and Mamba [14], have showcased remarkable proficiency in sequence modeling. These models rely on a classical continuous system that maps a one-dimensional input function or sequence, denoted as $x(t) \in \mathcal{R}$, through intermediate implicit states $h(t) \in \mathcal{R}^N$ to an output $y(t) \in \mathcal{R}$. The aforementioned process can be formulated as a linear Ordinary Differential Equation (ODE):

$$\begin{aligned} h'(t) &= \mathbf{A}h(t) + \mathbf{B}x(t), \\ y(t) &= \mathbf{C}h(t), \end{aligned} \quad (1)$$

where $\mathbf{A} \in \mathcal{R}^{N \times N}$ denotes the evolution matrix, while $\mathbf{B} \in \mathcal{R}^{N \times 1}$ and $\mathbf{C} \in \mathcal{R}^{N \times 1}$ represents the projection parameters.

S4 and Mamba discretize this continuous system, making it more suitable for deep learning scenarios. Specifically, they introduce a timescale parameter $\Delta$ and apply fixed discretization rules to transform $\mathbf{A}$ and $\mathbf{B}$ into discrete parameters $\bar{\mathbf{A}}$ and $\bar{\mathbf{B}}$. Typically, zero-order hold (ZOH) is employed as the discretization rule, which can be defined as follows

$$\begin{aligned} \bar{\mathbf{A}} &= \exp(\Delta \mathbf{A}), \\ \bar{\mathbf{B}} &= (\Delta \mathbf{A})^{-1}(\exp(\Delta \mathbf{A}) - \mathbf{I}) \cdot \Delta \mathbf{B}. \end{aligned} \quad (2)$$

After discretization, SSM-based models can be computed through linear recurrence, described as

$$\begin{aligned} h'(t) &= \bar{\mathbf{A}}h(t) + \bar{\mathbf{B}}x(t), \\ y(t) &= \mathbf{C}h(t). \end{aligned} \quad (3)$$

Another computational approach is global convolution, which can be formulated as follows

$$\begin{aligned} \bar{\mathbf{K}} &= (\mathbf{C}\bar{\mathbf{B}}, \mathbf{C}\bar{\mathbf{A}}\bar{\mathbf{B}}, \cdots, \mathbf{C}\bar{\mathbf{A}}^{L-1}\bar{\mathbf{B}}), \\ y &= \mathbf{x} * \bar{\mathbf{K}}, \end{aligned} \quad (4)$$

where $\bar{\mathbf{K}} \in \mathcal{R}^L$ represents a structured convolutional kernel, and $L$ denotes the length of the input sequence $x$.

## 4 THE PROPOSED METHOD

In this section, we initially introduce the formation of motion generation from sparse tracking, i.e., the positional and rotational data from a headset and two handheld controllers. Subsequently, the detailed architecture of the proposed MMD is described.

### 4.1 Problem Formation

Consider a sequence of human motion from sparse tracking $\mathbf{c} \in \mathbb{R}^{L \times C}$, where $L$ denotes the length of the sequence and $C$ represents the dimensions of the observed joint features. Our objective is to estimate the corresponding full-body poses $\mathbf{x} \in \mathbb{R}^{L \times J}$ based on $\mathbf{c}$, where $J$ denotes the dimensions of the full-body joint features. To represent human poses, we utilize the SMPL model parameters [29] and follow the practice of disregarding joints on the hands and face [10, 12, 24].

In our approach, we view the full-body pose estimation task as a conditional generation problem, where the sparse tracking serves as the conditioner. Specifically, a simple yet effective diffusion model detailed in Section 4.2 is proposed for motion generation. In the forward diffusion process, we gradually add Gaussian noise into

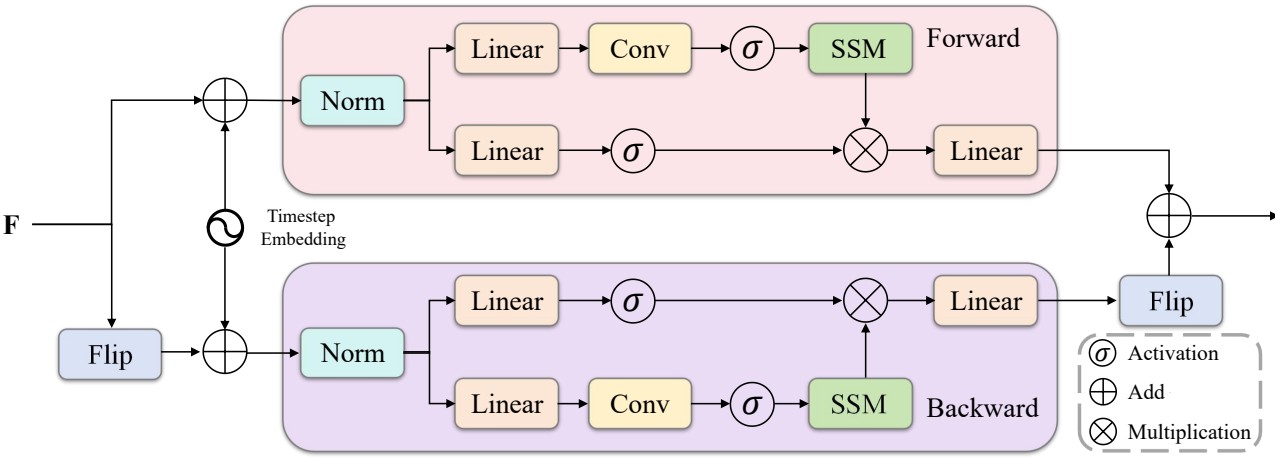

**Figure 3: The detailed architecture of the Temporal Mamba Module. As a key component in the network, it adopts a bidirectional strategy to fully capture the temporal dependencies inherent in the motion sequences.**

the clean data until the output distribution is close to an isotropic Gaussian distribution. Taking a data pair $(\mathbf{x}_0, \mathbf{c})$ in the dataset as a instance, the diffusion process from clean data $\mathbf{x}_0$ to Gaussian noise $\mathbf{x}_T$ is defined as

$$q\left(\mathbf{x}_1, \cdots, \mathbf{x}_T \mid \mathbf{x}_0\right) = \prod_{t=1}^{T} q\left(\mathbf{x}_t \mid \mathbf{x}_{t-1}\right),$$

$$q\left(\mathbf{x}_t \mid \mathbf{x}_{t-1}\right) = \mathcal{N}\left(\mathbf{x}_t; \sqrt{\alpha_t}\mathbf{x}_{t-1}, (1-\alpha_t)\,\mathbf{I}\right), \quad (5)$$

where $\mathbf{x}_t \in \mathbb{R}^{L \times J}$ represents the motion at the $t$-th step and there are $T$ denoising steps in total. $\mathbf{x}_T$ tends to an isotropic Gaussian distribution when $T \rightarrow \infty$. The hyperparameters $\alpha_t \in (0, 1)$ are predefined positive constants and should gradually decrease to 0 at later steps.

In the reverse diffusion process, the sampled Gaussian noise will be iteratively denoised to reconstruct a clean human motion sequence conditioned on the sparse tracking inputs $\mathbf{c}$. To obtain the clean data $\mathbf{x}_0$, we learn a generative process $p_\theta$ to predict and eliminate the noise added in the forward diffusion process. Specifically, the reverse process from Gaussian noise $\mathbf{x}_T$ to clean data $\mathbf{x}_0$ is defined as

$$p_\theta\left(\mathbf{x}_0, \cdots, \mathbf{x}_{T-1} \mid \mathbf{x}_T, \mathbf{c}\right) = \prod_{t=1}^{T} p_\theta\left(\mathbf{x}_{t-1} \mid \mathbf{x}_t, \mathbf{c}\right),$$

$$p_\theta\left(\mathbf{x}_{t-1} \mid \mathbf{x}_t, \mathbf{c}\right) = \mathcal{N}\left(\mathbf{x}_{t-1}; \mu_\theta(\mathbf{x}_t, \mathbf{c}, t), \sigma_t^2 \mathbf{I}\right), \quad (6)$$

where $\mu_\theta(\mathbf{x}_t, \mathbf{c}, t)$ is a neural network parameterized by $\theta$, which could be reformulated as

$$\mu_\theta(\mathbf{x}_t, \mathbf{c}, t) = \frac{1}{\sqrt{\alpha_t}}\left(\mathbf{x}_t - \frac{1-\alpha_t}{\sqrt{1-\bar{\alpha}_t}}\epsilon_\theta(\mathbf{x}_t, \mathbf{c}, t)\right), \quad (7)$$

and the variance $\sigma_t^2$ is a timestep dependent constant. The parametrization is $\sigma_t^2 = \frac{1-\bar{\alpha}_{t-1}}{1-\bar{\alpha}_t}(1-\alpha_t)$, where $\bar{\alpha}_t = \prod_{i=1}^{t} \alpha_i$. Therefore, the model has to learn to predict the residual noise $\epsilon_\theta(\mathbf{x}_t, \mathbf{c}, t)$. In our case, we follow the practice of [41], directly predicting the clean body

poses $\mathbf{x}_0$ instead of the residual noise $\epsilon_\theta$. The objective function in training can be formulated as

$$\mathcal{L}(\theta) = \mathbb{E}_{\mathbf{x}_0 \sim p_{\text{motion}}}\left[\|\mathbf{x}_0 - \hat{\mathbf{x}}_0\|_2^2\right], \quad (8)$$

where $p_{\text{motion}}$ is the distribution of the full-body poses in the dataset and $\mathbf{x}_0$ denotes the output of the proposed diffusion model.

## 4.2 Motion Mamba Diffusion

As shown in Figure 2, our Motion Mamba Diffusion (MMD) adopts a U-shaped architecture to process motion information across different time scales. This architecture is constructed using a meticulously designed Motion Mamba Block (MMB), which consists of two key components: Temporal Mamba Module (TMM) and Spatial Mamba Module (SMM). In particular, TMM specializes in modeling temporal dependencies among frames, enabling a thorough comprehension of the dynamics present in motion sequences. Meanwhile, SMM is employed to enhance the motion features within a single frame, which is dedicated to improving the understanding of the human body structure within each frame. The proposed components in our framework are introduced in detail as follows.

**Feature Mapping**: At time step $t$, fully connected layers are first applied to map the noisy human poses $\mathbf{x}_t \in \mathbb{R}^{L \times J}$ and the observed joint features $\mathbf{c} \in \mathbb{R}^{L \times C}$ to a unified latent space. Then we can obtain the corresponding latent features $\mathbf{f}_x \in \mathbb{R}^{L \times D}$ and sparse features $\mathbf{f}_c \in \mathbb{R}^{L \times D}$, where $D$ represents the dimension of the latent space. Subsequently, $\mathbf{f}_x$ and $\mathbf{f}_c$ are concatenated and input into a MLP layer for feature enhancement [12]. We denote the output motion features as $\mathbf{f} \in \mathbb{R}^{L \times D}$. Notably, the diffusion time step $t$ is also transformed into an embedding vector $\mathbf{f}_t \in \mathbb{R}^{1 \times D}$, which will be injected repetitively into the network.

**Temporal Mamba Module**: To capture the temporal dependencies present in motion sequences, we propose the Temporal Mamba Module (TMM). It is worth noting that we employ a bidirectional strategy to enhance its temporal modeling capabilities, as shown in Figure 3.

---

**Algorithm 1** Temporal Mamba Module Process

---

**Input:** compressed features $\mathbf{F}$ : $(B, L, D)$, timestep embedding $\mathbf{T}$ : $(B, 1, D)$
**Output:** transformed features $\mathbf{O}$ : $(B, L, D)$
 1: /* broadcast the timestep embedding $\mathbf{T}$ */
 2: $\mathbf{T}^*$ : $(B, L, D) \leftarrow \mathbf{Broadcast}(\mathbf{T})$
 3: /* bidirectional process */
 4: **for** $k$ in {forward, backward} **do**
 5:     $\mathbf{F}'$ : $(B, L, D) \leftarrow \mathbf{Flip}(\mathbf{F})$ if $k$ is backward **else** $\mathbf{F}' \leftarrow \mathbf{F}$
 6:     /* add the timestep embedding $\mathbf{T}$ */
 7:     $\mathbf{F}'$ : $(B, L, D) \leftarrow \mathbf{F}' + \mathbf{T}^*$
 8:     /* normalize the input features $\mathbf{F}'$ */
 9:     $\mathbf{F}^*$ : $(B, L, D) \leftarrow \mathbf{Norm}(\mathbf{F}')$
10:     $\mathbf{x}$ : $(B, L, E) \leftarrow \mathbf{Linear^x}(\mathbf{F}^*)$
11:     $\mathbf{z}$ : $(B, L, E) \leftarrow \mathbf{Linear^z}(\mathbf{F}^*)$
12:     $\mathbf{x}'_o$ : $(B, L, E) \leftarrow \mathbf{SiLU}(\mathbf{Conv1d}_o(\mathbf{x}))$
13:     $\mathbf{B}_o$ : $(B, L, N) \leftarrow \mathbf{Linear}_o^B(\mathbf{x}'_o)$
14:     $\mathbf{C}_o$ : $(B, L, N) \leftarrow \mathbf{Linear}_o^C(\mathbf{x}'_o)$
15:     /* softplus ensures positive $\Delta_o$ */
16:     $\Delta_o$ : $(B, L, E) \leftarrow \log(1 + \exp(\mathbf{Linear}_o^\Delta(\mathbf{x}'_o) + \mathbf{Parameter}_o^\Delta))$
17:     /* shape of $\mathbf{Parameter}_o^A$ is $(E, N)$ */
18:     $\bar{\mathbf{A}}_o$ : $(B, L, E, N) \leftarrow \Delta_o \bigotimes \mathbf{Parameter}_o^A$
19:     $\bar{\mathbf{B}}_o$ : $(B, L, E, N) \leftarrow \Delta_o \bigotimes \mathbf{B}_o$
20:     $\mathbf{y}$ : $(B, L, E) \leftarrow \mathbf{SSM}(\bar{\mathbf{A}}_o, \bar{\mathbf{B}}_o, \mathbf{C}_o)(\mathbf{x}'_o)$
21:     $\mathbf{y}'$ : $(B, L, D) \leftarrow \mathbf{Linear}(\mathbf{y} \odot \mathbf{SiLU}(\mathbf{z}))$
22:     $\mathbf{y}_o$ : $(B, L, D) \leftarrow \mathbf{Flip}(\mathbf{y}')$ if $k$ is backward **else** $\mathbf{y}_o \leftarrow \mathbf{y}'$
23: **end for**
24: /* sum and return*/
25: $\mathbf{O}$ : $(B, L, D) \leftarrow \mathbf{y}_{forward} + \mathbf{y}_{backward}$ **Return:** $\mathbf{O}$

---

**Algorithm 2** Spatial Mamba Module Process

---

**Input:** compressed features $\mathbf{O}$ : $(B, L, D)$, timestep embedding $\mathbf{T}$ : $(B, 1, D)$
**Output:** transformed features $\mathbf{P}$ : $(B, L, D)$
 1: /* transpose the compressed features $\mathbf{O}$ */
 2: $\mathbf{O}_t$ : $(B, D, L) \leftarrow \mathbf{Transpose}(\mathbf{O})$
 3: /* transform and broadcast the timestep embedding $\mathbf{T}$ */
 4: $\mathbf{T}'$ : $(B, 1, L) \leftarrow \mathbf{Linear}(\mathbf{T})$
 5: $\mathbf{T}^*$ : $(B, D, L) \leftarrow \mathbf{Broadcast}(\mathbf{T}')$
 6: /* bidirectional process */
 7: **for** $k$ in {forward, backward} **do**
 8:     $\mathbf{O}'$ : $(B, D, L) \leftarrow \mathbf{Flip}(\mathbf{O}_t)$ if $k$ is backward **else** $\mathbf{O}' \leftarrow \mathbf{O}_t$
 9:     /* add the timestep embedding $\mathbf{T}$ */
10:     $\mathbf{O}'$ : $(B, D, L) \leftarrow \mathbf{O}' + \mathbf{T}^*$
11:     /* normalize the input features $\mathbf{O}'$ */
12:     $\mathbf{O}^*$ : $(B, D, L) \leftarrow \mathbf{Norm}(\mathbf{O}')$
13:     $\mathbf{x}$ : $(B, D, E) \leftarrow \mathbf{Linear^x}(\mathbf{O}^*)$
14:     $\mathbf{z}$ : $(B, D, E) \leftarrow \mathbf{Linear^z}(\mathbf{O}^*)$
15:     $\mathbf{x}'_o$ : $(B, D, E) \leftarrow \mathbf{SiLU}(\mathbf{Conv1d}_o(\mathbf{x}))$
16:     $\mathbf{B}_o$ : $(B, D, N) \leftarrow \mathbf{Linear}_o^B(\mathbf{x}'_o)$
17:     $\mathbf{C}_o$ : $(B, D, N) \leftarrow \mathbf{Linear}_o^C(\mathbf{x}'_o)$
18:     /* softplus ensures positive $\Delta_o$ */
19:     $\Delta_o$ : $(B, D, E) \leftarrow \log(1 + \exp(\mathbf{Linear}_o^\Delta(\mathbf{x}'_o) + \mathbf{Parameter}_o^\Delta))$
20:     /* shape of $\mathbf{Parameter}_o^A$ is $(E, N)$ */
21:     $\bar{\mathbf{A}}_o$ : $(B, D, E, N) \leftarrow \Delta_o \bigotimes \mathbf{Parameter}_o^A$
22:     $\bar{\mathbf{B}}_o$ : $(B, D, E, N) \leftarrow \Delta_o \bigotimes \mathbf{B}_o$
23:     $\mathbf{y}$ : $(B, D, E) \leftarrow \mathbf{SSM}(\bar{\mathbf{A}}_o, \bar{\mathbf{B}}_o, \mathbf{C}_o)(\mathbf{x}'_o)$
24:     $\mathbf{y}'$ : $(B, D, L) \leftarrow \mathbf{Linear}(\mathbf{y} \odot \mathbf{SiLU}(\mathbf{z}))$
25:     $\mathbf{y}^*$ : $(B, D, L) \leftarrow \mathbf{Flip}(\mathbf{y}')$ if $k$ is backward **else** $\mathbf{y}^* \leftarrow \mathbf{y}'$
26:     $\mathbf{y}_o$ : $(B, L, D) \leftarrow \mathbf{Transpose}(\mathbf{y}^*)$
27: **end for**
28: /* sum and return*/
29: $\mathbf{P}$ : $(B, D, L) \leftarrow \mathbf{y}_{forward} + \mathbf{y}_{backward}$ **Return:** $\mathbf{P}$

---

In particular, we present the operations of TMM in Algorithm. 1. To operate over an input feature sequence $\mathbf{F} \in \mathbb{R}^{B \times L \times D}$ of batch size $B$, a timestep embedding $\mathbf{T} \in \mathbb{R}^{B \times 1 \times D}$ will be first broadcast and added to guide the denoising process (For backward, we will first flip $\mathbf{F}$ along the dimension of the sequence length). Then a normalization layer is applied to normalize $\mathbf{F}$. Next, we linearly project the normalized sequence to $\mathbf{x}$ and $\mathbf{z}$ with dimension size $E$. Subsequently, the 1-D convolution is applied to the $\mathbf{x}$ to obtain the $\mathbf{x}'_o$. Then we linearly project the $\mathbf{x}'_o$ to the $\mathbf{B}_o$, $\mathbf{C}_o$, $\Delta_o$, respectively. $\Delta_o$ is then utilized to transform $\bar{\mathbf{A}}_o$, $\bar{\mathbf{B}}_o$, respectively. In the end, we compute $\mathbf{y}_{\text{forward}}$ and $\mathbf{y}_{\text{backward}}$ with the SSM (For backward, we will flip the output along the dimension of the sequence length). The $\mathbf{y}_{\text{forward}}$ and $\mathbf{y}_{\text{backward}}$ will be added together to form the final output sequence $\mathbf{O} \in \mathbb{R}^{B \times L \times D}$. Note that $E$ represents the expanded state dimension, while $N$ denotes the SSM dimension.

**Spatial Mamba Module**: This module is introduced to learn the spatial dependencies of the motion poses within a single frame. Similar to TMM, the bidirectional strategy is adopted to improve the comprehension of the human body structures. It is worth noting that SMM is structurally consistent with TMM. The difference is that it process features along the dimension of the latent space rather than the sequence length.

We present the process of TMM in Algorithm. 2. Specifically, given the output $\mathbf{O}$ of TMM, we first transpose it along the last two dimensions to get the $\mathbf{O}_t \in \mathbb{R}^{B \times D \times L}$. Correspondingly, we linearly project the timestep embedding $\mathbf{T}$ to $\mathbf{T}'$ with dimension size $L$ and then broadcast it. The subsequent calculation process follows a similar procedure to TMM. Note that we utilize the transpose operation in the end to ensure the shape consistency between the output $\mathbf{P}$ and the input $\mathbf{O}$.

**Training**: At timestep $t$, we denote the output motion feature of our MMD as $\mathbf{F}_t$. It will be passed through a fully connected layer to obtain the full-body motion $\mathbf{x}_t$, which serves as the input for the next time step. This process will continue until we obtain the clean human poses $\mathbf{x}_0$. In training, the loss function adopted by our MMD is calculated in the same way as Equation 8.

## 5 EXPERIMENTS

### 5.1 Datasets

To verify the effectiveness of our method, we conduct extensive experiments on three subsets of AMASS [31]: CMU [1], BMLr [45], and HDM05 [34] for training and testing. The AMASS dataset is a large human motion database, which unifies different existing optical marker-based MoCap datasets and represents human poses with SMPL model parameters [29]. Following [24], we split the three datasets into training and test sets with 90% and 10% of the data, respectively.

### 5.2 Metrics

Following previous works, the adopted metrics for evaluation can be divided into three categories: 1) **Rotation-Oriented Metrics**: Mean Per Joint Rotation Error (MPJRE) in degrees measures the

**Table 1: Performance comparison with state-of-the-art approaches on AMASS.**

| Method | MPJRE ↓ | MPJPE ↓ | MPJVE ↓ | Hand PE ↓ | Upper PE ↓ | Lower PE ↓ | Root PE ↓ | Jitter ↓ | Upper Jitter ↓ | Lower Jitter ↓ |
|---|---|---|---|---|---|---|---|---|---|---|
| LoBSTr [49] | 10.69 | 9.02 | 44.97 | - | - | - | - | - | - | - |
| CoolMoves [2] | 5.20 | 7.83 | 100.54 | - | - | - | - | - | - | - |
| VAE-HMD [10] | 4.11 | 6.83 | 37.99 | - | - | - | - | - | - | - |
| AvatarPoser [24] | 3.08 | 4.18 | 27.70 | 2.12 | 1.81 | 7.59 | 3.34 | 14.49 | 7.36 | 24.81 |
| DAP [9] | 2.69 | 3.68 | 24.03 | - | - | - | - | - | - | - |
| AGRoL-MLP [12] | 2.69 | 3.93 | 22.85 | 2.62 | 1.89 | 6.88 | 3.35 | 13.01 | 9.13 | 18.61 |
| AGRoL [12] | 2.66 | 3.71 | 18.59 | 1.31 | 1.55 | 6.84 | 3.36 | 7.26 | 5.88 | 9.27 |
| MMD (Ours) | **2.30** | **3.17** | **17.32** | **0.79** | **1.25** | **5.94** | **2.86** | **6.52** | **4.89** | **8.87** |
| Ground Truth | 0 | 0 | 0 | 0 | 0 | 0 | 0 | 4.00 | 3.65 | 4.52 |

**Table 2: Results of cross-dataset evaluation among different methods. The best results for each dataset and each evaluation metrics are highlighted in boldface. Note that AGRoL is not included in this comparison due to its absence of cross-dataset experiments.**

| Dataset | Method | MPJRE ↓ | MPJPE ↓ | MPJVE ↓ |
|---|---|---|---|---|
| CMU | CoolMoves [2] | 9.20 | 18.77 | 139.17 |
| | LoBSTr [49] | 12.51 | 12.96 | 49.94 |
| | VAE-HMD [10] | 6.53 | 13.04 | 51.69 |
| | AvatarPoser [24] | 5.93 | 8.37 | 35.76 |
| | DAP [9] | 5.46 | 8.15 | 32.32 |
| | MMD(Ours) | **5.26** | **7.88** | **28.25** |
| BMLrub | CoolMoves [2] | 7.93 | 13.30 | 134.77 |
| | LoBSTr [49] | 10.79 | 11.00 | 60.74 |
| | VAE-HMD [10] | 5.34 | 9.69 | 51.80 |
| | AvatarPoser [24] | 4.92 | 7.04 | 43.70 |
| | DAP [9] | 4.75 | 6.81 | 42.78 |
| | MMD(Ours) | **4.47** | **5.68** | **26.17** |
| HDM05 | CoolMoves [2] | 9.47 | 17.90 | 140.61 |
| | LoBSTr [49] | 13.17 | 11.94 | 48.26 |
| | VAE-HMD [10] | 6.45 | 10.21 | 40.07 |
| | AvatarPoser [24] | 6.39 | 8.05 | 30.85 |
| | DAP [9] | 6.18 | 7.84 | 29.17 |
| | MMD(Ours) | **5.84** | **7.37** | **27.33** |

average relative rotation error for all joints; 2) **Velocity-Oriented Metrics**: Mean Per Joint Velocity Error (MPJVE) in cm/s evaluates the average velocity error for all joints, while Jitter in $10^2\text{m/s}^3$ quantifies the mean jerk; and 3) **Position-Oriented Metrics**: Mean Per Joint Position Error (MPJPE) in cm measures the average position error across all joints, while Root PE, Hand PE, and Upper and Lower PE estimate the corresponding parts of human body, respectively.

## 5.3 Implementation Details

Due to its simplicity and continuity, we represent the joint rotations by the 6D reparametrization. Therefore, for the observed joint features $c \in \mathbb{R}^{N \times C}$, which consists of the orientation, translation, orientation velocity and translation velocity of the head and hands in global coordinate system, $C = 18 \times 3$. For the sequences of full-body poses $x \in \mathbb{R}^{N \times D}$, $D = 22 \times 6$. Unless otherwise stated, we set the length of motion sequence $N$ to 196 and the sampling steps to 8 for inference. All experiments were conducted using the PyTorch platform [36] and trained/tested on a single NVIDIA A100 GPU. In addition, our network used Adam optimizer [26] with a base learning rate of 3e-4.

## 5.4 Comparison with State-of-the-Art Methods

We compare Motion Mamba Diffusion (MMD) with the state-of-the-art method (AGRoL) and other popular body tracking models. The quantitative experimental results are tabulated in Table 1, where the best results are boldfaced. It can be seen that our proposed method showcases significant advancements, achieving superior performance across all metrics and outperforming the current state-of-the-art technique. In particular, three key metrics from the AMASS dataset demonstrate MMD's superiority: a reduction in MPJRE by 13.53%, MPJPE by 14.55%, and MPJVE by 6.83%. Additionally, we measure the performance corresponding to different parts of the body. Specifically, MMD reduce the position error of the hands, upper body, lower body, and root by 39.69%, 19.35%, 13.16%, and 14.88%, respectively.

Moreover, it is worth noting that our method exhibits a significant 22.70% improvement in Jitter, indicating its capability to generate smoother human motion, a crucial aspect for enhancing user experience and realism. The improvement in upper body and lower body is 44.39% and 8.42%, respectively.

To further validate the performance of our proposed method, we conduct a 3-fold cross-dataset evaluation among different approaches. Specifically, we train our model on two subsets and then test on the remaining subset in a round-robin fashion. Table 2 presents the experimental results of various methods evaluated on the CMU, BMLrub, and HDM05 datasets. Our method achieves the best results across all evaluation metrics in all three datasets. Particularly noteworthy is our MMD, which significantly outperforms previous state-of-the-art methods in terms of MPJVE across all subsets, indicating smoother motions. Additionally, both the rotation error (MPJRE) and position error (MPJPE), representing prediction accuracy, are notably reduced.

For visualization, we conducted qualitative experiments and the results are shown in Figure 4, in which the extent of the error is indicated by the intensity in red. The results show that MMD

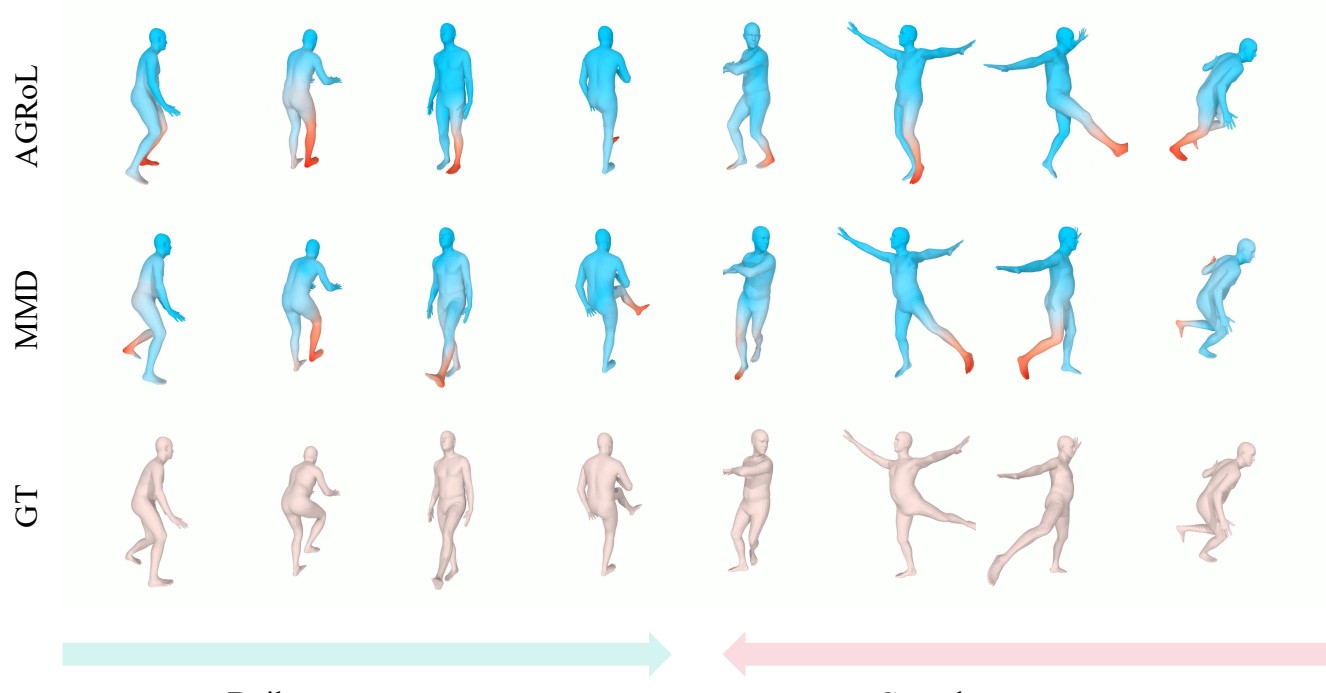

**Figure 4: Visual comparison between the AGRoL and our MMD on static movements. The extent of the error is indicated by the intensity in red. The first row denotes the results of the state-of-the-art model AGRoL. The second row represents the reconstruction results of our models. The third row is the ground truth of full body.**

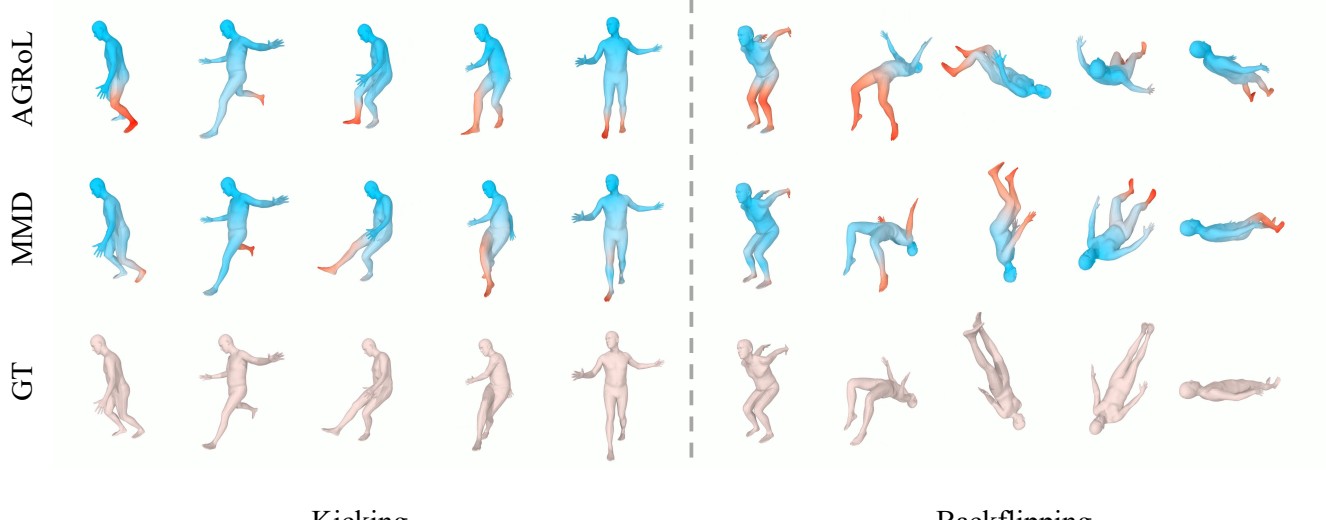

**Figure 5: Visual comparison between the AGRoL and our MMD on dynamic movements. Compared to AGRoL [12], our MMD can generate more accurate and smoother motion sequences.**

generates more accurate human motion compared with AGRoL. For complex movements, our method can also produce plausible full-body predictions from merely three sparse tracking inputs. Additionally, two different motion sequences are presented for further comparison. As shown in Figure 5, the results generated

from our method exhibit higher smoothness and better consistency with the ground truth.

In addition to delivering high-quality predictions, our MMD exhibits exceptional efficiency. It achieves an average processing time of only 0.57ms per pose when running on an A100 GPU, resulting

in an impressive frame rate of 1754 frames per second (fps). This remarkable performance makes our method ideal for real-time avatar animation driven by head-mounted displays (HMDs) in practical applications.

## 5.5    Ablation Studies

In this section, we present extensive ablation studies on our Motion Mamba Diffusion (MMD) architecture, aiming to assess the impact and effectiveness of each proposed component on the overall model performance. The specific details are as follows.

   **The effectiveness of proposed modules.** A series of ablation experiments were conducted to verify the effectiveness of our proposed components. As can be seen in Table 3, the individual application of the Temporal Mamba Module (TMM) and Spatial Mamba Module (SMM) brings significant performance improvements, while the combination of the two modules can lead to better results. Notably, the first line represents that only the forward SSM is applied in the TMM, excluding the SMM, while the second line denotes that we utilize the bidirectional TMM.

**Table 3: Ablation results on two key components: Temporal Mamba Module (TMM) and Spatial Mamba Module (SMM), revealing their impact on MPJRE, MPJPE, and MPJVE metrics.**

| TMM | SMM | MPJRE ↓ | MPJPE ↓ | MPJVE ↓ |
|:---:|:---:|:---:|:---:|:---:|
|  |  | 2.68 | 3.78 | 25.10 |
| ✓ |  | 2.49 | 3.37 | 18.92 |
|  | ✓ | 2.52 | 3.48 | 18.84 |
| ✓ | ✓ | **2.46** | **3.31** | **18.07** |

   **The number of blocks in the framework**. We investigate the influence of varying the number of Motion Mamba Blocks (MMBs) in our MMD network. The number of MMBs in Figure 2 is denoted as $[E_1, E_2, M, D_1, D_2]$ from left to right, respectively. As shown in Table 4, the model's performance consistently improves as more blocks are added. Nevertheless, the performance gains start to plateau when too many blocks are used.

**Table 4: Ablation study on the number of blocks in the proposed Motion Mamba Diffusion (MMD) framework.**

| # Blocks | MPJRE ↓ | MPJPE ↓ | MPJVE ↓ |
|:---:|:---:|:---:|:---:|
| [3, 1, 1, 1, 3] | 2.54 | 3.47 | 18.89 |
| [3, 2, 2, 2, 3] | 2.50 | 3.42 | 18.51 |
| [4, 2, 2, 2, 4] | **2.46** | **3.31** | **18.07** |
| [5, 2, 2, 2, 5] | 2.48 | 3.33 | 18.13 |
| [5, 3, 3, 3, 5] | 2.48 | 3.32 | 18.08 |

   **Input Sequence Length**. Our method takes a sequence of sparse tracking signals as input and predicts the full-body poses of the same length. In this ablation study, we evaluate the performance of

five different input sizes: 64, 96, 144, 196, and 256. The results are summarized in Table 5. As shown in Table 5, the model achieves the best overall performance across three key metrics when the sequence length is set to 96. Therefore, we opt for this length for subsequent experiments.

**Table 5: Ablation study on the input & output length of the MMD network. The best overall performance is achieved when we set the length to 96.**

| # Sequence Length | MPJRE ↓ | MPJPE ↓ | MPJVE ↓ |
|:---:|:---:|:---:|:---:|
| 64 | **2.26** | **3.06** | 18.23 |
| 96 | 2.28 | 3.11 | **17.63** |
| 144 | 2.42 | 3.26 | 17.68 |
| 196 | 2.46 | 3.31 | 18.07 |
| 256 | 2.69 | 3.63 | 18.23 |

   **The number of sampling steps for inference**. In Table 6, we ablate the number of sampling steps during inference. It is worth noting that even when evaluating with merely 2 sampling steps, our model can achieve plausible performance. Meanwhile, we notice that the model tends to output worse results when too many sampling steps are used. To achieve the best overall performance, 5 is selected as our final sampling steps for inference.

**Table 6: Ablation study on the number of sampling steps for inference. The results become worse when the number of sampling steps is too large.**

| # Sampling Steps | MPJRE ↓ | MPJPE ↓ | MPJVE ↓ |
|:---:|:---:|:---:|:---:|
| 2 | 2.77 | 4.21 | 18.06 |
| 5 | 2.30 | 3.17 | **17.32** |
| 8 | **2.28** | **3.11** | 17.63 |
| 50 | 2.32 | 3.15 | 19.01 |
| 100 | 2.33 | 3.16 | 19.28 |
| 1000 | 2.36 | 3.20 | 19.76 |

## 6    CONCLUSION

In this paper, we propose Motion Mamba Diffusion (MMD), a novel SSM-based framework for accurately estimating full-body motion from sparse tracking signals. It is a pioneering method, which effectively leverages the robust generation capabilities of diffusion models in conjunction with the sequence modeling proficiency of SSMs to generate realistic human poses. Specifically, we adopt a U-shaped architecture and introduce two innovative modules, Temporal Mamba Module (TMM) and Spatial Mamba Module (SMM), to model the temporal motion sequence and human body structure within each frame, respectively. Extensive experiments demonstrate that our method outperforms the latest approaches and achieves new state-of-the-art performance, showcasing its significant potential in various generative multimedia applications.

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
