# OpenReview forum: "Realistic Full-Body Motion Generation from Sparse Tracking with State Space Model"
_acmmm.org/ACMMM/2024/Conference — MM2024 Oral_

### Official Review · Reviewer_F2pB · 2024-05-19

**Rating:** 4
**Confidence:** 2

**Summary:**

This paper proposes a motion Mamba diffusion model to estimate the corresponding full-body posture based on sparse tracking. This model utilizes diffusion models for motion generation and state space models for sequence modeling. This model generates motion through a motion Mamba block, which includes a bidirectional time Mamba module (TMM) to recognize temporal dependencies between frames and a spatial Mamba module (SMM) to enhance single-frame motion features.

**Strengths:**

1. This paper addresses the problem of sequence modeling and motion generation.
2. This paper can generate realistic full-body movements.

**Limitations:**

1. What is the difference between the proposed Mamba diffusion model and the general diffusion model? Is it simply introducing Mamba into the diffusion model?
2. What are the specific details of the proposed motion Mamba model?
3. The expression of SMM in line 516 of 4.2 is incorrect.
4. In the experimental section, which method did the data in lines 671 and 674 decrease compared to? It is not evident from the table data that the data described in the paper has been reduced. Where is the ambiguity of the data in lines 677 and 680 reflected?

**Suitability:**

2

---

### Official Review · Reviewer_eH67 · 2024-05-24

**Rating:** 4
**Confidence:** 2

**Summary:**

This paper studies full-body motion generation from sparse tracking. A state space model-based framework, named Motion Mamba Diffusion, is proposed to generate human poses by iterative denoising. This framework comprises two components: a Temporal Mamba Module to capture temporal dependencies inherent in motion sequences and a Spatial Mamba Module to enhance features within each frame. Experiments on the AMASS dataset show that the proposed method achieves good performance and runs in real time.

**Strengths:**

- This paper is well-written and easy to follow.

- The proposed model performs better than the previous methods on the AMASS dataset.

**Limitations:**

- Failure cases and limitation of the proposed method should be analyzed.

- This paper proposes bidirectional Temporal/Spatial Mamba Modules for motion generation. The related paper [1] also proposes a Hierarchical Temporal Mamba Block and a Bidirectional Spatial Mamba Block for motion generation. While [1] generates motion from text, this paper generates motion from sparse tracking. However, more discussions and comparisons of the two methods should be provided.


[1] Motion Mamba: Efficient and Long Sequence Motion Generation with Hierarchical and Bidirectional Selective SSM, arXiv preprint arXiv:2403.07487 (2024).

**Suitability:**

3

---

### Official Review · Reviewer_J5P9 · 2024-05-25

**Rating:** 4
**Confidence:** 4

**Summary:**

This paper introduces a SSM-based framework that enhanced the accuracy of full body motion generation from sparse tracking. The framework introduces two mamba module for modeling temporal relations and spatial relations.

**Strengths:**

1: The introduction of new structure Mamba increases the accuracy of full body motion generation from sparse tracking.

2: The paper is well-structured and easy to read.

**Limitations:**

1:Missing citation on motion generation from sparse tracking, especially works including spatial and temporal modeling, considering the contribution of this work. A lot of related works in this area should be included and analysized in related work:

Xiaozheng Zheng, Zhuo Su, Chao Wen, Zhou Xue, Xiaojie Jin. 2023. Realistic Full-Body Tracking from Sparse Observations via Joint-Level Modeling. In Proceedings of the IEEE/CVF International Conference on Computer Vision. 14678-14688

Feiyu Yao, Zongkai Wu, and Li Yi. 2024. Full-Body Motion Reconstruction with Sparse Sensing from Graph Perspective. In Proceedings of the AAAI Conference on Artificial Intelligence, Vol. 38. 6612-6620.

Jose Luis Ponton, Haoran Yun, Andreas Aristidou, Carlos Andujar, and Nuria Pelechano. 2023. SparsePoser: Real-time Full-body Motion Reconstruction from Sparse Data. ACM Transactions on Graphics (TOG) 43, 1 (2023), 1–14.

Jiaman Li, C. Karen Liu, Jiajun Wu. 2023. Ego-Body Pose Estimation via Ego-Head Pose Estimation. In Proceedings of the IEEE/CVF Conference on Computer Vision and Pattern Recognition, 17142–17151.

2: More analysis is needed to prove the mamba modolue effectiveness.
(1) The speed is slower. AGRoL produces 196 output frames in 35 ms on V100 (16/32GB), which is 0.17ms per pose, while this paper achieves 0.57ms per pose on A100 (40/80GB). Is this worth to trade computing time for precision improvement? A Comparison of computational efficiency is persuasive.
(2) Performance comparison on three subsets of AMASS is not enough. Related works with diffusion or VAE structrure (such as FLAG, AGRoL and HuMoR) use nearly all subsets of AMASS and even other dataset (AIST++) to prove the effectiveness of the method. Adding more experiments is persuasive.

I will change my rating to Accept if limitations 1 and limitations 2(2) are solved.

**Suitability:**

3

---

### Meta-Review · Area_Chair_yMrL · 2024-07-09

**Recommendation:** Accept (Oral)
**Confidence:** 5

**Metareview:**

This paper propose a novel state space approach to estimate human body pose from sparse tracking signals exploiting diffusion models and MAMBA. The experimental results are extensive. An interesting extension of this work should regard the evaluation of actual generative capability of the model. The rebuttal satisfied the reviewers which in general raised their scores.